# Survey, Detection, Characterization of Papaya Ringspot Virus from Southern India and Management of Papaya Ringspot Disease

**DOI:** 10.3390/pathogens12060824

**Published:** 2023-06-11

**Authors:** Udavatha Premchand, Raghavendra K. Mesta, Venkatappa Devappa, Mantapla Puttappa Basavarajappa, Venkataravanappa Venkataravanappa, Lakshminarayana Reddy C. Narasimha Reddy, Kodegandlu Subbanna Shankarappa

**Affiliations:** 1Department of Plant Pathology, College of Horticulture, University of Horticultural Sciences, Bagalkot 587104, India; chandpremu04@gmail.com (U.P.); basump@rediffmail.com (M.P.B.); ksshankarappa@gmail.com (K.S.S.); 2Division Crop Protection, ICAR-Indian Institute of Horticultural Research, Bengaluru 560090, India; 3Department of Plant Pathology, College of Agriculture, University of Agricultural Sciences, GKVK, Bengaluru 560065, India

**Keywords:** papaya ringspot virus, disease incidence, characterization, recombination, complete genome, integrated disease management, insecticide, biorationals, seaweed extract, modules

## Abstract

Papaya ringspot virus (PRSV) is a significant threat to global papaya cultivation, causing ringspot disease, and it belongs to the species *Papaya ringspot virus*, genus *Potyvirus*, and family *Potyviridae.* This study aimed to assess the occurrence and severity of papaya ringspot disease (PRSD) in major papaya-growing districts of Karnataka, India, from 2019 to 2021. The incidence of disease in the surveyed districts ranged from 50.5 to 100.0 percent, exhibiting typical PRSV symptoms. 74 PRSV infected samples were tested using specific primers in RT-PCR, confirming the presence of the virus. The complete genome sequence of a representative isolate (PRSV-BGK: OL677454) was determined, showing the highest nucleotide identity (nt) (95.8%) with the PRSV-HYD (KP743981) isolate from Telangana, India. It also shared an amino acid (aa) identity (96.5%) with the PRSV-Pune VC (MF405299) isolate from Maharashtra, India. Based on phylogenetic and species demarcation criteria, the PRSV-BGK isolate was considered a variant of the reported species and designated as PRSV-[IN:Kar:Bgk:Pap:21]. Furthermore, recombination analysis revealed four unique recombination breakpoint events in the genomic region, except for the region from HC-Pro to VPg, which is highly conserved. Interestingly, more recombination events were detected within the first 1710 nt, suggesting that the 5’ UTR and P1 regions play an essential role in shaping the PRSV genome. To manage PRSD, a field experiment was conducted over two seasons, testing various treatments, including insecticides, biorationals, and a seaweed extract with micronutrients, alone or in combination. The best treatment involved eight sprays of insecticides and micronutrients at 30-day intervals, resulting in no PRSD incidence up to 180 days after transplanting (DAT). This treatment also exhibited superior growth, yield, and yield parameters, with the highest cost–benefit ratio (1:3.54) and net return. Furthermore, a module comprising 12 sprays of insecticides and micronutrients at 20-day intervals proved to be the most effective in reducing disease incidence and enhancing plant growth, flowering, and fruiting attributes, resulting in a maximized yield of 192.56 t/ha.

## 1. Introduction

Papaya (*Carica papaya* L.) belongs to the family *Caricaceae*, is a fast-growing, short-lived plant grown in tropical and subtropical regions of the world for its fruit, papain, pectin, and antibacterial properties [1,2]. Members of the *Caricaceae* family originated in Africa, where two species still exist today. Papaya is the third-most cultivated tropical crop in the world. Brazil and India are the largest producers, although Mexico is the main exporting country [3]. Among common fruits, papaya ranks first on nutritional scores in terms of the percentage of vitamin A, vitamin C, potassium, folate, niacin, thiamine, riboflavin, iron, calcium, and fiber [4]. The global production of papaya is about 13.74 MT and is grown in an area of 4.62 lakh ha [5]. India is the largest producer, accounting for nearly 44.04 percent of global papaya production (6.05 MT) with an area of 1.49 lakh ha, and it has emerged as an alternative cash crop after banana.

Pests and diseases are major threats to papaya production worldwide. Of these, fungal and viral diseases are of global significance, causing serious damage to fruit production and devastating the entire crop [6]. Among the viral diseases, papaya ringspot and leaf curl diseases are more prevalent [7,8]. Globally, papaya ringspot disease (PRSD) is the most devastating disease affecting papaya production in almost every region where it is grown [9], which is a major constraint for the papaya industry. While, papaya leaf curl disease (PLCD) associated with begomoviruses is prevalent in Asian countries, recent reports indicate that begomovirus infecting papaya has been observed in American countries as well, but not in a widespread manner [9,10]. The PRSV infected plants exhibit symptoms of mosaic and chlorosis on the leaf lamina, water-soaked lesions on the petioles and upper part of the trunk, and distortion of young leaves, which results in the formation of shoestring symptoms. Papaya plants infected at the early vegetative stage may remain stunted, sometimes become bunchy top and never produce any fruit, leading to 100 percent yield loss [11,12]. Whereas in plants infected at the reproductive stage, entire leaves become yellow and ‘ring spots’ will appear on the fruits, leading to yield losses of 85.0 to 90.0 percent [13,14]. In addition, PRSV is naturally transmitted by the aphids in a nonpersistent manner [15] and mechanically transmitted under laboratory conditions [16]. PRSV belongs to the genus *Potyvirus*, of the family *Potyviridae* [17]. There are two predominant pathotypes in PRSV, namely PRSV-P, which infects both papaya and cucurbits, and PRSV-W, which infects only members of the *Cucurbitaceae* family [18]. However, under experimental conditions, PRSV-P isolates infect members of the families *Caricaceae*, *Cucurbitaceae*, and *Chenopodiaceae*. Whereas, PRSV-W isolates infect plants of the families *Cucurbitaceae* and *Chenopodiaceae* [19]. The virions are nonenveloped flexuous filamentous particles (760–800 nm × 12 nm) with a single-stranded monopartite (+) RNA genome encapsidated by coat protein (CP), and viral particles contain 94.5 percent protein and 5.5 percent nucleic acid. The length of the PRSV genome is ~10,326 nt, having the 5’ end and a poly-A tail at the 3’ terminus, which encodes a large polypeptide of 3344 amino acids (~350 kDa) [16,17,20]. The polypeptide is proteolytically cleaved into 10 final products: P1 (Protein 1), HC-Pro (Helper Component-Protease), P3 (Protein 3), 6K1 (6-kDa peptide 1), CI (Cylindrical Inclusion protein), 6K2 (6-kDa peptide 2), VPg (Viral Protein genome-linked), NIa-Pro (Nuclear inclusion A-protease), NIb (Nuclear inclusion B), and CP [21]. Papaya ringspot disease (PRSD) was first reported from the island of Ohau in Hawaii state by Parris [22], which was well known for papaya production, and it was named “Wailu’ disease by Linder [23]. Later, the name ‘papaya ringspot virus’ was coined by Jensen [24]. Then, onward the distribution and incidence of PRSD, which is caused by PRSV was reported from almost all the countries in the world [17,18]. In India, PRSD was first reported by Capoor and Verma [25], and later it was reported from many geographical regions of India [26].

Many promising approaches are available globally to manage plant viral diseases, the safest among them being biological control and the use of virus-resistant crop lines [27,28,29]. Another common and effective approach is the control of virus vectors [30,31]. The use of resistant varieties/hybrids is the only sustainable, economical, and eco-friendly method to contain PRSV. However, attempts have also been made to control PRSV through a transgenic approach; primarily involving resistance mediated by the replicase and coat protein (CP) genes via post-transcriptional gene silencing (PTGS) [32,33,34,35]. Nevertheless, the breakdown of single or double virus resistance in CP gene transgenic papaya has been observed under field conditions due to more virulent PRSV strains 5 to 19 [36]. Therefore, currently, no papaya variety exhibits absolute resistance to PRSV naturally. Disease management strategies also include immunization to obtain resistant varieties and prophylactic measures to restrain viral dispersion. Unfortunately, in the absence of a natural PRSV resistant cultivar or hybrid [37], it is important to go for prophylactic measures (pesticides) which can significantly reduce disease incidence. However, indiscriminate use of insecticides results in the evolution of insecticide-resistant insect population and exerts negative effects on the environment [38]. Therefore, biorational and biostimulant-based management methods have emerged as alternatives to pesticide use, gaining importance in recent decades [39,40,41,42,43,44,45,46,47]. These methods indirectly manage PRSV by controlling the vectors [48]. Biorational products such as neem oil, pongamia oil, groundnut oil, and mineral oil, as well as biostimulants such as seaweed, have been found to be effective as insect repellents or in controlling insect vectors, thereby significantly delaying or reducing the incidence of PRSV [47,48,49,50,51,52]. Thus, they can be utilized as components in an integrated disease management approach. However, only a few attempts have been made to evaluate these biorationals and biostimulants as management practices against insect pests in controlling viral diseases [53]. Hence, considering the research work conducted so far and the existing research gap, the present research has been undertaken to study the incidence of the disease, the molecular characterization of the virus, and the development of management strategies to reduce the incidence of PRSD. 

## 2. Materials and Methods

### 2.1. Survey and Collection of Virus Isolates

An intensive survey was carried out during the year 2019–2020 and 2020–2021 in 10 major papaya growing districts (Bagalkote, Belagavi, Ballary, Chitradurga, Gadag, Haveri, Kalaburagi, Koppal, Vijayapura, and Yadgiri) of Karnataka, India to determine the incidence and distribution of PRSD. The disease incidence was estimated by counting the number of infected plants visually by the total number of plants in a selected plot size of 25 × 25 sq. m area in each field. The percent disease incidence (PDI) was calculated by using the following formula.
$$\mathrm{PDI}\;(\%)=\frac{\mathrm{Number}\;\mathrm{of}\;\mathrm{infected}\;\mathrm{plants}}{\mathrm{Total}\;\mathrm{number}\;\mathrm{of}\;\mathrm{plants}\;\mathrm{observed}}\times 100$$

Meanwhile, during the survey, 74 symptomatic leaf samples exhibiting various kinds of symptoms were collected from naturally infected papaya plants from each surveyed location. These samples were collected separately in polythene bags, labeled, and brought immediately to the Plant Pathology Laboratory, College of Horticulture, Bagalkote, University of Horticultural Sciences (UHS), Bagalkot, India, snap-frozen in liquid nitrogen and stored at −80 °C for further detection and characterization of the virus. 

### 2.2. Detection and Sequencing of PRSV

To confirm the presence of viral infection in 74 symptomatic papaya leaf samples collected during the survey, a pathogenicity test was conducted. This test involved the sap inoculation of collected samples onto a healthy papaya cultivar, ‘Red Lady’ using 0.1 M phosphate buffer with a pH of 7.5. Additionally, to detect PRSV in these samples, total RNA was extracted using the Spectrum™ Plant Total RNA Kit from Sigma-Aldrich (Catalogue No. STRN50; St. Louis, MO, USA). The RNA was quantified using Thermo Fisher Scientific Nanodrop TM 2000 spectrophotometer and diluted to 1000 ng/µL. Total RNA extracted from 74 samples was subjected to cDNA synthesis using PrimeScript™ 1st strand cDNA Synthesis Kit (Takara Bio, Catalog No. #6110A; Kusatsu, Japan) following the manufacturer’s instructions in a thermal cycler (Eppendorf Mastercycler: nexus gradient). Subsequently, synthesized cDNA was used for PCR amplification using KAPA Taq PCR Reagent (KK1016-500 U) with a set of PRSV CP gene specific primers (MB 11A/MB 11B) [54] (Appendix A). Additionally, screening for mixed-virus infection associations under natural field conditions was performed on all confirmed PRSV surveyed samples using virus-specific primers (Appendix A) to detect viruses such as papaya milk vetch dwarf virus [55], papaya leaf distortion virus [56], papaya mosaic virus [56], and zucchini yellow mosaic virus [57].

The complete genome sequence of the PRSV-BGK isolate was determined using nine pairs of overlapping primers [58]. The resulting PCR amplicons of desired length were purified using NucleoSpin^®^ Gel and a PCR Clean-up kit (Takara Bio, Catalog No. # 740986.50). Purified PCR fragments were cloned into the plasmid vector pMD20-T (Takara Bio Mighty TA-cloning Kit, Takara Bio Catalogue No. #6028). Subsequently, each recombinant clone was transformed into the DH5α strain of *Escherichia coli* [59] and successfully transformed colonies were identified by colony PCR [58]. The colonies that showed positive for transformation were cultured and used for plasmid isolation. Plasmid DNA was isolated from each transformed colony by using the alkaline lysis method, and the presence of the insert was confirmed by restriction digestion with the BamH1 restriction enzyme. Nine confirmed clones for the presence of PRSV genome fragments were sequenced using an automated DNA sequencer ABI (Applied Biosystems, Waltham, MA, USA) at Medauxin Pvt. Ltd., Bengaluru, Karnataka, India.

### 2.3. PRSV Complete Genome Comparison, Phylogenetic, and Recombination Analysis

VecScreen software was used to remove vector sequences from the PRSV genome sequences obtained. A complete aligned sequence of PRSV was obtained from the genome sequence fragments using BioEdit (version 7.2) (http://bioedit.software.informer.com) (accessed on 16 February 2023) [60]. The aligned sequence of the PRSV-BGK isolate was used for sequence similarity search in the NCBI database using BLASTn (http://www.ncbi.nlm.nih.gov) (accessed on 16 February 2023). Sequences of PRSV isolates available in GenBank showing the maximum blast score with the complete genome sequence of PRSV-BGK isolate were retrieved and used for analysis (Appendix A). Pairwise sequence identity analysis was carried out using the Sequence Demarcation Tool (SDT) version 1.2 with default parameters to obtain the color-coded matrix of the identity scores. The nt and aa identities of the complete genome sequence of the PRSV-BGK isolate (OL677454) were compared with the complete genome sequences of PRSV isolates reported in GenBank from different parts of the world. Multiple sequence alignments were conducted using the ClustalW algorithm [61], and open reading frames (ORFs) in the sequence were identified using the NCBI ORF finder (http://www.ncbi.nlm.nih.gov/gorf/gorf.html) (accessed on 17 February 2023). A dendrogram was constructed for both the nt as well as the aa sequence of PRSV using the neighbor-joining method by MEGA11 with a 60 percent cut-off value and with 1000 bootstrap replicates [62]. Recombination breakpoint analysis between the PRSV-BGK isolate and retrieved PRSV isolates was carried out using RDP 5 by employing Bonferroni correction with a confidence value of greater than 95 percent (*p*-value 0.05) [63]. To ensure reliability, the PRSV-BGK sequence was considered recombinant if the recombination signal was supported by at least three methods. In RDP analysis, the length of the window was set to 10 variable sites, and the step size was set to one nucleotide. *p*-values were estimated by randomizing the alignment 1000 times.

### 2.4. Integrated Management of PRSV Disease under Field Conditions

Field experiments were conducted over two consecutive years (2019–2020 and 2020–2021) to assess the effect of insecticides and biorationals (plant-based oils and seaweed extract) against PRSD at Haveli experimental station, UHS, Bagalkot, Karnataka, India (16°12′07.0′′ N 75°41′08.9′′ E). Experiments were conducted in a randomized block design with three replications and nine different treatments (schedule of eight foliar sprays at every 30-day intervals), details of which are given in Appendix A. Susceptible papaya cultivar ‘Red Lady’ was planted with a spacing of 1.5 m × 1.5 m and followed all the agronomical practices as per the recommended package of practices (POP) of the UHS, Bagalkot, Karnataka, India [64] to raise the crop in good conditions, except for the usage of insecticides and biorational sprays. Further, using the effective treatment(s) from the above experiment(s), three integrated disease management (IDM) modules (each module had a schedule of 12 foliar sprays at 20-day intervals) were designed for the management of PRSD on papaya. These three modules were evaluated along with the recommended dose of POP of the UHS, Bagalkot, Karnataka, India [64] as a check under natural disease pressure during the year 2020–21 at Mannikeri village, Bagalkote district, Karnataka, India (16°19′07.3′′ N, 75°38′42.6′′ E). Each experimental plot was 10.5 m × 13.5 m in size and planted with the cultivar ‘Red Lady’ at a spacing of 1.5 m × 1.5 m, containing 72 plants per module (Appendix A).

Observations on disease incidence (%) were recorded visually along with plant height (cm), plant girth (cm), number of leaves per plant, inter nodal length (cm), days taken for first flowering, number of flowers per plant, number of days from flowering to first fruit set, number of days from fruit set to harvest, fruit length (cm), fruit breadth (cm), fruit diameter (cm), cavity diameter (cm), yield per plant (kg), and yield per hectare (t) were recorded at every 30-day intervals, and collected data from the different variables were analyzed separately in the ANOVA to show the significant differences for any particular effect using OPSTAT (http://14.232.166/opstat/) (accessed on 10 February 2023) online software [65]. The net returns, cost–benefit ratio (B:C), and incremental cost–benefit ratio (ICBR) of the experiment were calculated as below to determine the economic feasibility of adapting the results to the farmer’s field.
Net returns (INR) = Total returns − Total cost



$$B:C=\frac{\mathrm{Total}\;\mathrm{returns}\;(\mathrm{INR}/\mathrm{ha})}{\mathrm{Total}\;\cos t\;(\mathrm{INR}/\mathrm{ha})}$$


$$\mathrm{ICBR}=\frac{\mathrm{Additional}\;\mathrm{returns}\;\mathrm{over}\;\mathrm{control}\;(\mathrm{INR}/\mathrm{ha})}{\mathrm{Additional}\;\cos t\;\mathrm{over}\;\mathrm{control}\;(\mathrm{INR}/\mathrm{ha})}$$



## 3. Results

### 3.1. Survey, Incidence, and Symptomatology of PRSD

A total of 74 surveyed farmers’ fields in 10 major papaya growing districts of Karnataka, India, recorded the incidence of PRSD. The average disease incidence ranged from 50.5 to 100 percent in the surveyed districts. The highest average incidence of PRSD was recorded from the districts of Koppal (100.0%) and Yadgir (100.0%), followed by Haveri (91.5%), Bagalkote (85.8%), Ballari (70.3%), Belagavi (80.8%), Vijayapura (67.0%), and Chitradurga (61.9%). The minimum average incidence of disease was recorded from Gadag (50.5%) and Kalaburagi (57.3%) districts (Appendix A and Appendix A). Typical symptoms of PRSD were recorded in the infected papaya plants during the survey, such as green mosaic, yellow mosaic, leaf curling, ring spots, stunted growth, puckering of leaves, mottling, bunch top, blistering, distortion of leaves, shoestring, oily streak, chlorotic spots, and distorted fruit. Of these, green mosaic was present at all the locations, irrespective of the age of the plants and surveyed locations (Figure 1). While surveying, it was observed that younger plants (below five months) at the prevegetative stage recorded only green mosaic symptoms. However, symptoms such as a yellow mosaic, leaf curling, stunted growth, puckering, mottling, blistering on leaves, and shoestring symptoms were common in plants in the prereproductive stage (above five months old plants). Further, more severe symptoms were recorded at the reproductive stage of the crops. Distortion of leaves was also seen in severely infected plants.

### 3.2. Molecular Characterization of PRSV

PRSV infection was detected in all 74 samples collected from different farmers’ fields through a pathogenicity test (Appendix A) and also through RT-PCR, using a set of CP gene-specific primers (MB 11A/MB 11B) [54], which resulted in the expected PCR amplicon of ~905 nt. However, none of the samples showed positive for mixed-viral infections, confirming that no other viral infections were associated with papaya under natural field conditions in the major surveyed locations in Karnataka, India.

Nucleotide sequence analysis of the amplified CP gene amplicons from 74 papaya samples revealed that they have more than 97 percent identity among themselves. Therefore, one representative sample (PRSV-BGK) collected from Bagalkote, Karnataka, India, was used for complete genome characterization using a pair of nine overlapping primers designed by Ortiz-Rojas and Chaves-Bedoya [58]. The genome amplicons obtained were cloned and sequenced. The consensus complete genome sequence obtained was deposited in NCBI, GenBank under the accession number, OL677454. The complete genome of the PRSV-BGK isolate is 10,341 nt long and consists of a single large ORF of 10,029 nt commencing at the position of 86th nt and terminating with UGA at position between 10,112nd–10,114th nt, which encodes a polyprotein of 3342 aa residues with a protein weight of 380.24 kDa. It contains 10 mature peptides residing on a virion (+ve) sense strand in a clockwise direction, followed by 85 nt (1st–85th nt) and 227 nt (10,115th–10,341st nt) at 5′ and 3′ untranslated regions (UTR) with a conserved terminal sequence, i.e., 5′-AAATAAAACATCT and -CTCTTAGAATGAG-3′, respectively (Table 1 and Figure 2).

The nt position of the 86th–1723rd codes for the P1 ORF, which contains well-characterized conserved motifs, such as VELI (431st–434th aa), GSSG (496th–499th aa), and RG (520th–522nd aa) which may have a role in the protease activity. Additionally, the MEQY/N motif (546th–547th aa) serves as the internal cleavage point for the protease to cleave P1 and HC-Pro. HC-Pro, located at 1724th–3094th nt, contains important motifs involved in aphid transmission (KITC at 597th–600th aa and PTK at 855th–857th aa positions), protease activity (NIFLAML at 892nd–898th aa and AELPRILVDH at 947th–956th aa positions), and cell-to-cell long-distance movement of the virions (LKANTV motif at 977th–982nd aa positions). The HYIVG/G motif (1003rd–1004th aa) serves as the cleavage site between the HC-Pro and P3 regions. Similarly, P3 is located at the 3095th–4129th nt position in the genome and contains an aa cleavage motif VIHQ/A (1348th–1349th aa) between P3 and 6K1 regions. 6K1, located at 4130th–4285th nt, is the shortest region of the genome, with only 156 nt. It terminates at the VYHQ/S (1400th–1401st aa) motif between the 6K1 and CI regions. CI (4286th–6190th nt) contains the AVGSGKST motif (1486th–1493rd aa) region for helicase function, and the CI/6K2 region cleaves at the VYHQ/G (2032nd–2035th aa) region. The 6K2 ORF is located in the genome position of 6191st–6361st nt and further cleaves at VFHQ/G (2089th–2092nd aa) between the 6K2 and VPg regions. The VPg coding region is present between 6362nd–6928th nt, which cleaves at the motif of VHHE/G between VPg and the NIa-Pro region. The NIa-Pro (6929th–7642nd nt) having the conserved motif VFEQ/S (2516th–2519th aa) at the interaction of the NIa-Pro and NIb regions. Finally, in the region of NIb (7643rd–9253rd nt), CP (9254th–10,114th nt) sequences showed the conservation of important motifs such as GDD in the NIb region for replicase activity, DAG in the CP region for aphid transmission, and GNNSGQPSTVVDNTLMV (2825th–2841st aa) and NGDDL (2868th–2872nd aa) for RNA-dependent RNA polymerase activity. The cleavage site at the VYHQ/S (3053rd–3056th aa) motif present at the intersection of the NIb and CP regions was also conserved.

### 3.3. Comparison of the Complete Genome of PRSV-BGK Isolate with Other PRSV Sequences

Analysis showed that the PRSV-BGK isolate (OL677454) had an nt identity of 70.2 to 95.8 percent with other Indian (12) and worldwide (46) PRSV isolates available in the NCBI database. The maximum nt identity of 95.8 percent was with the PRSV-HYD isolate (KP743981) from Hyderabad, Telangana, India, which is geographically close to Karnataka. This was followed by the PRSV-Pune VC isolate (MF405299) at 95.5 percent and the PRSV-Pune PS3-H isolate (MF405297) at 93.9 percent, both reported from Maharashtra, India. The lowest sequence identity was 70.2 percent with PRSV BD-1 (MH444652 reported from Bangladesh) (Appendix A and Appendix A).

At the aa level, the PRSV-BGK (OL677454) isolate exhibited an aa identity of 83.1 to 96.5 percent with PRSV isolates from different parts of the world. Maximum aa identities of 96.5 percent and 96.2 percent with PRSV-Pune VC isolate (MF405299) reported from Maharashtra, India, and PRSV-HYD isolate (KP743981) from Hyderabad, Telangana, India, respectively, which are both geographically close to Karnataka. Further, the lowest aa identity of 83.1 percent with PRSV BD-1 (MH444652) was reported from Bangladesh (Appendix A and Figure 3b). Based on the species demarcation criteria for the complete ORF given by the International Committee on Taxonomy of Viruses (ICTV), the PRSV-BGK isolate (OL677454) is demarcated as a variant of an already reported species. Further, based on sampling location and host, it has been given a descriptor as papaya ringspot virus-[India:Karnataka:Bagalkote:Papaya:2021] and designated as PRSV-[IN:Kar:Bgk:Pap:21].

### 3.4. Phylogenetic Analysis of PRSV

Complete nucleotide genome sequences of 58 PRSV isolates reported worldwide were used for phylogenetic analysis along with a PRSV-BGK isolate (OL677454). PRSV isolates were found to be clustered into two major lineages (I-Lineage and II-Lineage), one lineage having a mixture of isolates from different parts of the world, whereas another lineage had four monophyletic groups based on regions, viz., Asian, American, Mexican, and Indian. The I-Lineage (mixed) contains a total of 12 isolates [Bangladesh (2), Papua New Guinea (5), China (1), Mexico (1), and India (3)]. In II-Lineage, the Asian-clade contains 21 isolates of Asian origin including two Indian isolates (KJ755852 and MF356497). However, the American-clade contains 11 isolates except for two isolates (X67673 and NC001785, Taiwan origin), all of which are American origin. Whereas four isolates of Mexican origin and a single isolate of Papua New Guinea origin are grouped into a separate clade. In the Indian origin phylogroup (Indian-clade), the majority of PRSV isolates originated from the Indian subcontinent (MF405297, MF405295, MF405296, MF405298, EF017707, KP743981, and MF405299) and formed a single and separate phylogroup. However, the PRSV-BGK (OL677454) isolate formed a distinct branch with the PRSV-HYD (KP743981) isolate infecting papaya reported from Hyderabad, India. Further, the phylogenic analysis also showed that the Indian subcontinent clade has a distinct relationship with Mexican isolates (Appendix A).

The second phylogenetic analysis of PRSV isolates is based on the deduced aa of the polyprotein, in which PRSV isolates are divided into two lineages (Figure 3a); Mexican lineage had only five isolates of Mexican origin. Whereas another lineage is composed of five clusters, viz., East-Asia, American-Oceana, America, South-Asia, and India. There were three subclusters within the clusters, and the sole member is present in the subcluster, i.e., MH311882, MF356497, and MH444652. The present isolate (PRSV-BGK:OL677454) falls in the Indian-clade with seven other India isolates, including PRSV-HYD (KP743981), with which it is closely clustered, even when the nt sequences were analyzed.

### 3.5. Recombination Analysis of PRSV

Recombination analysis detected four unique recombination breakpoint events in the genomic region of PRSV-BGK (OL677454), except for the regions of HC-Pro, P3, 6K1, CI, 6K2, and VPg, which are highly conserved (Figure 4). More recombination events were detected in the first 1710 nt of the PRSV genome, suggesting that the 5′ UTR and P1 regions are playing an important role in shaping the PRSV genome. The first recombination event spanned the region from 1st to 1031st nt (3′UTR to P1) and 10,328th to 10,341st (5′ UTR) with PRSV-Mild-Los-Rios (MT747167) as the major parent and PRSV-Umi (MF356497) as the minor parent (*p* < 1.026 × 10^−6^), respectively. Whereas major parent PRSV-HYD (KP743981) and minor parent PRSV-PM-I (MF405296) were detected to be the potential parents in two independent recombination events recorded at 1373rd to 1710th nt (P1) (*p* < 3.407 × 10^−2^) and 7327th to 9061st nt (NIa-Pro to NIb) (*p* < 6.155 × 10^−12^) positions and finally a fourth recombination event was detected between the 9205th and 10,075th nt (NIb to CP) region with major parent PRSV-TXG (NC_001785) and minor parent PRSV-PRSVR3 (KJ755852) (*p* < 5.576 × 10^−3^) (Table 2).

### 3.6. Integrated Management of PRSD under Field Conditions

#### 3.6.1. Effect of Insecticides and Biorationals on PRSD Incidence, Growth, and Yield 

##### Parameters of Papaya

Data on the effects of different insecticides (tolfenpyrad, imidacloprid, thiacloprid, and dinotefuran), biorationals (neem oil, pongamia oil, groundnut oil, and mineral oil), and a seaweed extract, along with micronutrients, either alone or in combination, were assessed for PRSD incidence, growth, and yield parameters of papaya plants at various growth stages (30, 60, 90, 120, 150, 180, 210, 240, 270, 300, 330, and 360 DAT) under natural disease pressure for two seasons, i.e., 2019–2020 and 2020–2021 (Appendix A). Since the results obtained in this experiment were consistent across both seasons, and papaya is a perennial in nature with long crop duration, it takes almost 12 months for the complete crop growth period. Considering this extended growth period, instead of conducting a third season, the data from the two seasons were pooled and analyzed statistically.

##### Disease Incidence

The results of the two-year field evaluation (pooled data) on the effect of insecticides and biorationals on disease incidence at different growth stages of papaya were significant (Table 3). The incidence of PRSD on ‘Red Lady’ cultivar of papaya increases as the crop grows, as shown in the heatmap in Figure 5 generated by R studio (R Studio Team, 2020). Disease incidence was nonsignificant (*p* > 0.05) between the experimental treatments at the initial (i.e., 30 and 60 DAT) and final (i.e., 360 DAT) crop growth stages. Initially, there was no incidence of PRSD observed. Whereas at the final crop growth stage, the incidence of PRSD in all the treatments is at its peak, i.e., 100 percent.

At the crop growth stage of 90 DAT, PRSD was recorded only in the control treatment (T_9_) with a 0.57 percent incidence, while the other treatments were free from the disease (Table 3). Further, significantly (*p* > 0.05), the maximum disease incidence at 120 DAT was recorded in T_9_ (65.61%) followed by T_8_ (10.63%), while no incidence was reported in the remaining treatments. At 150 DAT, the incidence was maximum in T_9_ (90.38%), followed by T_8_ (25.56%), and the least in T_6_ (3.65%) and T_2_ (4.0%), while other treatments were disease free. Plants attained the reproductive stage at 180 DAT, and all treatments had some level of disease incidence, except for T_1_ and T_5_. At this stage, the highest disease incidence (100%) was reached in the control treatment (T_9_), followed by T8 (55.56%), T6 (25.10%), T_2_ (19.65%), T_3_ (9.58%), T_4_ (2.61%), and T_7_ (2.19%). Whereas the first incidence was reported on T_1_ (1.49%) and T_5_ (10.97%) at only 210 DAT, by that stage the incidence on T_9_ (100%) and T_8_ (81.47%) was significantly higher (*p* > 0.05). This trend was consistent across all treatments at 240 DAT.

At 270 DAT, T_9_ and T_8_ had recorded the highest disease incidence (100%), which was significantly higher than the other treatments (*p* > 0.05). While T_1_ had the lowest incidence (24.07%), which was significantly lower than the other treatments such as T_2_ (79.39%), T_6_ (76.36%), T_3_ (74.07%), T_7_ (52.31%), T_4_ (46.28%), and T_5_ (39.81%). At 300 DAT, T_1_ witnessed only 50.90 percent disease incidence and was significantly (*p* > 0.05) the least among all treatments. Finally, when the plants reached their maximum growth stage at 330 DAT, all treatments except T_1_ (84.25%) had 100 percent disease incidence. These results indicate that the combination of four different insecticides (tolfenpyrad 15% EC, imidacloprid 17.8% SL, thiacloprid 21.7 SC, and dinotefuran 20% SG) and micronutrients in T_1_ was the most effective treatment for reducing the incidence of PRSD under field conditions.

##### Growth, Yield, and Yield Parameters

Results of the study showed that the effect of insecticides and biorationals on PRSD incidence was statistically significant (*p* > 0.05) for all the parameters. Table 4 shows that the T_1_ treatment had a significantly superior effect compared to the other treatments, including the control (T_9_), for all the growth and yield variables. The T_1_ treatment, which consisted of foliar spray of four different insecticides (tolfenpyrad 15% EC, imidacloprid 17.8% SL, thiacloprid 21.7 SC, and dinotefuran 20% SG) followed by micronutrients, recorded the highest plant height (225.93 cm), internodal length (4.11 cm), plant girth (40.45 cm), and number of leaves per plant (29.50). This was followed by T_5_ (216.71 cm, 4.10 cm, 39.68 cm, and 28.67, respectively) and T_4_ (214.18 cm, 4.10 cm, 39.48 cm, and 28.29, respectively), which were on par. While under control treatment, T_9_ recorded, significantly, the least plant height (135.53 cm), internodal length (3.28 cm), plant girth (20.95 cm), and number of leaves per plant (19.80) (Table 4). As observed in the case of percent incidence disease, a similar trend was observed in plant growth variables, where T_1_ was found to be effective in enhancing the growth of the plants. This enhanced growth is likely due to the lower disease incidence in the T_1_ treated plants.

Healthy crop growth has a positive effect on different yield and yield variables. This trend was also observed where T_1_ was significantly superior in the number of days taken for first flowering (93.01), number of flowers (73.06), number of days taken from flowering to first fruit set (94.01), and number of days taken from first fruit set to harvest (125.64) over other treatments. Further, T_5_ (95.94, 59.88, 96.38, and 123.88, respectively) and T_4_ (99.31, 56.70, 95.93, and 123.50, respectively) were found to be the next best ones; while the control treatment (T_9_) had the lowest performance among all the treatments. At the same time, the enhanced fruit parameters were also observed in T_1_, viz., number of fruits per plant (26.09), fruit diameter (13.13 cm), fruit length (27.78 cm), fruit breadth (14.43 cm), and fruit cavity diameter (6.81 cm), which are significantly superior, while the least fruit parameters were recorded in T_9_.

The incidence of disease disrupts the normal growth and physiological function of a plant. This will ultimately reflect in fruit yield. Treatment, T_9_ (control), presented the maximum incidence of the disease and recorded the least fruit yield per plant (0.83 kg) and fruit yield per hectare (2.57 t). However, T_1_ with the least incidence of diseases along with enhanced growth and yield parameters, resulted in a significantly excellent treatment concerning fruit yield per plant (57.86 kg) and fruit yield per hectare (178.56 t), which was followed by T_5_ (52.16 kg and 160.97 t, respectively) and T_3_ (45.83 kg and 141.44 t, respectively) (Figure 6 and Figure 7).

Further, analysis of the economic impact of the experiment for managing PRSD using insecticides and biorationals showed that the highest net returns were obtained from T_1_ (INR 7,69,064/ha), followed by T_5_ (INR 6,58,229/ha), T_3_ (INR 5,36,754/ha), and T_4_ (INR 4,83,757/ha). Similarly, the B:C ratio followed the same trend, i.e., 3.54 in T_1_, 3.14 in T_5_, 2.72 in T_3_, and 2.60 in T_4_. The ICBR was recorded highest in T_1_ (121), followed by T_4_ (88), T_5_ (68), and T_6_ (59) (Appendix A).

#### 3.6.2. Integrated Disease Management Modules for the Management of PRSD

The IDM modules were designed using the outcome of the above experiment to replicate its effective management strategies to manage the PRSD. Three designed modules (M_1_, M_2_, and M_3_) were tested along with POP of UHS, Bagalkot, Karnataka, India as a check (M_4_). Module M_1_ was designed by looking into the treatment T_1_, the most effective treatment in the above study, which consists of 12 foliar sprays of four different insecticides (tolfenpyrad 15% EC, imidacloprid 17.8% SL, thiacloprid 21.7 SC, dinotefuran 20% SG) followed by micronutrients. Module M_2_ is a combination of T_1_ and T_3_. Whereas the module M_3_ is a combination of T_1_, T_3_, and T_2_ treatments.

##### Disease Incidence

To see the effect of developed modules on PRSD, incidence was calculated to find out the effectiveness of each module. Disease incidence at initial growth stages, i.e., up to 120 DAT, showed no incidence of disease in any of the modules. This is comparable with the results of the above-management experiment. During 150 DAT, except for the M_1_ module, initiation of disease incidence was reported in the remaining three modules, i.e., M_4_ (2.76%), M_3_ (1.93%), and M_2_ (1.14%). When the plant reached its reproductive stage (i.e., 180 DAT), M_1_ (0.44%) recorded its foremost incidence, while the remaining modules were at moderate incidence levels, viz., M_2_ (10.72%), M_3_ (11.89%), and M_4_ (16.63%). Likewise, throughout the crop growth stages, M_1_ was at its lowest level of disease incidence, i.e., 210 (4.14%), 240 (20.11%), and 270 (56.44%) DAT and it took about 330 days (i.e., 330 DAT) to reach its maximum incidence (100%). While, other modules ended up with a peak (100%) level of disease incidence at 300 DAT. This outcome suggested that the designed M_1_ module was superior to others throughout crop growth stages (Appendix A).

##### Growth, Yield, and Yield Parameters

As observed in the case of disease incidence, a similar trend was observed in crop growth and yield variables, where M_1_ was found to be effective in enhancing the growth and yield of the plants. This enhanced growth and yield could be due to the lowest disease incidence on plants imposed by the M_1_ module (Appendix A). Likewise, the M_1_ module was found to have the maximum growth parameters, i.e., plant height (227.02 cm), internodal length (4.24 cm), plant girth (40.46 cm), and number of leaves per plant (29.19), followed by module M_2_. Similarly, the number of days taken for first flowering (92.78), number of flowers (73.38), number of days taken from flowering to first fruit set (94.63), and number of days taken from first fruit set to harvest (125.61) were recorded as superior (M_1_ module) over control (M_4_: recommended POP). Concerned fruit parameters, viz., number of fruits per plant (28.31), fruit diameter (14.23 cm), fruit length (28.00 cm), fruit breadth (14.63 cm), and fruit cavity diameter (7.14 cm) were also recorded at the highest level in the M_1_ module, along with fruit yield per plant (62.40 kg) and fruit yield per hectare (192.56 t), compared to control (M_4_: recommended POP).

## 4. Discussion

Worldwide, PRSV has been recognized as the most destructive viral pathogen on papaya [66]. Despite its importance, the national economies of many papaya-cultivated countries are threatened by the incidence of PRSD. The disease affects almost all stages of the crop and spreads very quickly to the whole orchard within three to seven months, which leads to yield losses of up to 100 percent [17,67,68]. Although PRSV occurs in different countries, higher levels of diversity were observed among Indian isolates compared to the rest of the world [69,70,71]. This might be due to a lack of resistant varieties, the fast evolution of the new strains of PRSV through recombination, and the occurrence of different aphid species [72,73]. In the current study, the incidence of PRSD was observed to range from 50.5 to 100.0 percent across different districts surveyed in Karnataka, India. Whereas more severe incidence of disease was recorded in the northern part of Karnataka.

Further, the most common commercial cultivars of papaya grown were ‘Red Lady’ and ‘Ice Burg’, which recorded 100 percent incidence in many surveyed areas and were found susceptible to PRSV infection. A similar observation was recorded by Pushpa et al. [47], where ‘Red Lady’, ‘Sunrise Solo’, and ‘Arka Surya’ were found highly susceptible to PRSV in south Karnataka, and Lokhande et al. [13] in the Vidharbha region of Maharashtra. These findings indicated that the disease is widespread in different parts of India, with varying levels of incidence. This may be due to growing susceptible cultivars such as ‘Red Lady’, ‘Sunrise Solo’, and ‘Arka Surya’ being grown in larger areas continuously, the variation in temperatures and relative humidity, which might have helped to build up the viral infection in papaya. The differences in the incidence of disease recorded in surveyed areas might also be due to the variation in the sources of inoculums, vector population, prevalent climatic conditions, and stage of crop plants [74].

The sampling survey conducted in the present study revealed that the incidence of PRSV was predominantly observed in all surveyed locations through a pathogenicity test. Furthermore, our preliminary RT-PCR analyses confirmed PRSV infection only, with no evidence of mixed-viral infection under natural field conditions in all the 74 surveyed samples collected from major papaya-growing locations in Karnataka, India. Amplified CP gene sequences revealed that the PRSV isolate associated with the samples is closely related to PRSV-HYD and PRSV-Pune VC isolates [7,75,76,77,78,79]. A similar trend was observed in the phylogenetic analysis of both nt and aa sequences, in which PRSV-BGK (OL677454) isolate was closely clustered with PRSV-HYD isolate (KP743981), an isolate from the Telangana state of India (geographical proximity). Similar types of phylogenies were also reported earlier [77,80].

Recombination provides a means by which viruses invade new hosts or develop greater pathogenicity and/or virulence [81,82]. The recombination analysis detected significant numbers of recombination breakpoints in the P1 region, followed by NIa, NIb, CP, 5′ UTR, and 3′ UTR regions in the current isolate. Whereas no recombination breakpoints were detected in HC-Pro, P3, 6K1, CI, 6K2, and VPg regions, which are highly conserved. The P1 region has a higher variability and recombination [7], suggesting that the 5′ UTR and P1 are playing an important role in shaping the PRSV genome [79]. Whereas no recombination event was detected in the 6K1 and 6K2 cistrons, which confirms earlier findings of Mangrauthia et al. [80] and Gorane et al. [79]. According to previous studies, it appears that recombination occurs repeatedly in the *Potyvirus* genus. Thus, recombination is one of the most important factors that allowed potyviruses to adapt to various hosts and different environments to survive and spread [83]. It seems that there might be some correlation between recombination, sequence pattern, and determining the origin of PRSV [80]. One of the major mechanisms in the evolution of PRSV is recombination, which may have a role in maintaining the effectiveness of purifying selection by preventing the accumulation of deleterious mutations. It has been proposed that recombination can be advantageous for RNA viruses, as it can create high-fitness genotypes more rapidly than mutation alone [84]. Recombination has been associated with the expansion of the host range, the modification of transmission vector specificity, the evolution of new viruses, an increase in virulence as well as pathogenesis, and the evasion of host immunity [85]. Recombination is an important factor in promoting virus evolution, which can increase genomic biodiversity [86] and enhance the virulence of the virus to expand its host range [87]. The diversity of PRSV varies from region to region due to the occurrence of spatial recombination and temporal evolution within the province [88]. PRSV diversity appears to be changing at different rates, presumably driven by introductions, movement of plant materials, geographical isolation, and disease management practices [89].

The current study on IDM revealed that T_1_ treatment (eight sprays of four different insecticides and micronutrients at 30 day-intervals) was proven to significantly reduce the incidence of PRSD at various crop growth stages and substantially enhance the growth and yield of the crop, along with a high-cost–benefit ratio of 1:3.54. Insecticides used in this treatment belong to the pyrazole (tolfenpyrad) and neonicotinoid (imidacloprid, thiacloprid, and dinotefuran) classes, which are widely used to manage insect pests. Nicotinic acetylcholine receptors (nAChRs) are the main target sites of neonicotinoid insecticides [90]. Whereas inhibition of complex I of the mitochondrial electron transport (respiratory) chain is the site of action for pyrazole [91], active mainly upon contact with egg, larval, nymphal, and adult stages of the pest [92]. Sprays of the pyrazole insecticide tolfenpyrad (15% EC) have been proven to be highly effective against sucking pests, with a safer effect on natural enemies in different agriculture and horticulture crops, viz., cotton [93], groundnut [94], cabbage [95,96,97,98], okra [99,100], cucumber [101], mango [102], fennel [103], and pomegranate [104], along with reducing the incidence of turnip mosaic virus (TuMV) disease in peach [105].

Neonicotinoids are the fastest growing class of insecticides in modern crop protection, with widespread use against a broad spectrum of sucking and certain chewing pests compared to conventional insecticides [106]. An earlier study showed that foliar application of dinotefuran (295 g a.i./ha) significantly reduced the cucurbit yellow stunting disorder virus (CYSDV) infecting cucurbits by controlling the disease transmitted by *Bemisia tabaci* [107]. Similarly, dinotefuran at 0.2 percent was also proven to be significantly effective in reducing the incidence of rice tungro disease infection up to 75.0 percent by controlling the insect vector population up to 94 percent [108]. On the other hand, imidacloprid (Confidor 20%) has been recorded to have a strong effect in reducing the PRSD incidence by up to 33.03 percent over the control, by effectively inhibiting the aphid population (86.33%) [109]. Further, the least mosaic incidence (5.2%) with a significant reduction in whitefly population (77.6%) was also recorded in brinjal when treated with thiacloprid 240 SC at 120 g ai/ha. The T_5_ treatment combination, i.e., eight sprays, a combination of four different insecticides and seaweed extract with micronutrients at 30-day intervals, where seaweed extract is sprayed at 2-month intervals to avoid residual toxicity due to continuous insecticidal sprays in the study, was found to be the second best treatment in controlling disease incidence along with good growth, flowering, and yield attributes.

Further, this treatment gave net returns of 6, 58,229 INR/ha with a high-cost–benefit ratio of 1:3.14. The effect of seaweed extract is well known for activating plants’ immune responses [110] by mimicking DAMP elicitors [111,112], along with increased disease resistance by significantly escalating the activities of glucanase, peroxidase, phenylalanine ammonia-lyase, chitinase, and polyphenol oxidase with higher levels of total phenols, and parallelly up regulating the marker genes involved in the jasmonic acid, salicylic acid, and ethylene-mediated defense pathways in seaweed treated crops [113]. The outcomes of various studies proved its inhibitory effect by inducing long-term protection against potato virus X (PVX) [114], tobacco mosaic virus [115], plum pox virus [116], and cucumber mosaic virus (CMV) [117], which upholds the current outcome.

Meanwhile, treatment T_3_ which contains eight sprays of only seaweed extract along with micronutrients, was recorded as free from PRSV incidence up to 150 DAT. Although the foremost incidence of disease was recorded during 180 DAT with 9.58 percent, its effect was not reflected too much on yield (141.44 t/ha), which may be due to diverse secondary metabolites including antimicrobial, antifungal, and antiviral compounds [118,119]. Concurrently, Pushpa et al. [47] conducted a foliar application of aqueous seaweed extract (*Kappaphycus alvarezii*) at 4 mL/L on papaya against PRSD at every 15-day interval and recorded the delay and relatively lesser percent disease incidence with fewer symptoms and well-formed fruits than the control plants. Moreover, plants were relatively taller with a dense foliar canopy and a higher average number of fruits (30/plant) than those in the untreated group (15/plant). However, despite growing evidence on the functions of diverse defense molecules present in seaweed extracts, their complex mode of action remains elusive [120]. Further studies in this direction are essential for a detailed understanding of their molecular mechanisms. Though a good yield was obtained in T_3_ treatment with seaweed extract, it is not encouraging to go for its recommendation at the field level as the cost–benefit ratio is much less (1:2.6) due to its higher cost. Seaweed extract may be incorporated judicially in the IDM module as one of the components.

The outcome of T_2_ (eight sprays of bio-oils with micronutrients) showed that complete dependence on biorationals is not advisable, as it recorded a higher rate of disease incidence with a lesser yield (80.79 t). However, these biorationals in combination with insecticide (T_4_) performed better, having less disease incidence and a 62.16 percent increase in yield. These biorationals can be integrated between the insecticidal sprays to lessen toxicity. Biorationals and chemicals have relatively postponed the rate of PRSV infection. This finding was encouraging, as an escape from early infection is reported to evade the severe reduction in yield [121]. However, a combination of reflective row cover, mineral oil, and imidacloprid spray was the most effective treatment in delaying the PRSV incidence (83.33% at 18 months after planting), while only mineral oil or neem oil application did not protect the crop [72]. Similarly, the different combinations of insecticides and oils differed in their effectiveness in delaying PRSV infection. A combination of neem oil 1% + dimethoate 1.05% recorded the least disease incidence (6.66% and 41.66%) at 60 and 150 DAP, respectively [48]. Further, results of the IDM module showed that Module I (12 sprays of four different insecticides and micronutrients at every 20-day interval) is the significantly best module as it recorded the least disease incidence while enhancing plant growth, flowering, and fruiting attributes, ultimately maximizing the yield (192.56 t/ha).

## 5. Conclusions

Our results indicate that PRSD is prevalent in almost all papaya growing areas of Karnataka State, India, irrespective of the age of the papaya crop. The PRSV isolate characterized in the present study is a closely related PRSV-HYD isolate (KP743981) from Hyderabad, Telangana, India. This is the first complete genomic characterization of PRSV from Karnataka (Southern India). The developed IDM strategy for two seasons proved that eight sprays of four different insecticides, i.e., tolfenpyrad 15% EC at 1 mL/L, imidacloprid 17.8% SL at 0.2 mL/L, thiacloprid 21.7 SC at 1 mL/L, and dinotefuran 20% SG at 0.5 g/L, alternatively with micronutrients at every 30-day interval, were found to be the most effective against PRSV, with no incidence of disease at early crop growth stages (up to 180 DAT), which is a critical period for PRSV infection, along with maximum yield and high-cost–benefit ratio (1:3.54). Hence, it can be considered the best approach for efficiently managing PRSD under field conditions. Furthermore, module (M_1_) which contains 12 sprays of insecticides, i.e., tolfenpyrad 15% EC at 1 mL/L, imidacloprid 17.8% SL at 0.2 mL/L, thiacloprid 21.7 SC at 1 mL/L, and dinotefuran 20% SG at 0.5 g/L, alternatively three times, and micronutrients at a 20-day interval, is established as the best IDM module for managing the PRSD.

## Figures and Tables

**Figure 1 pathogens-12-00824-f001:**
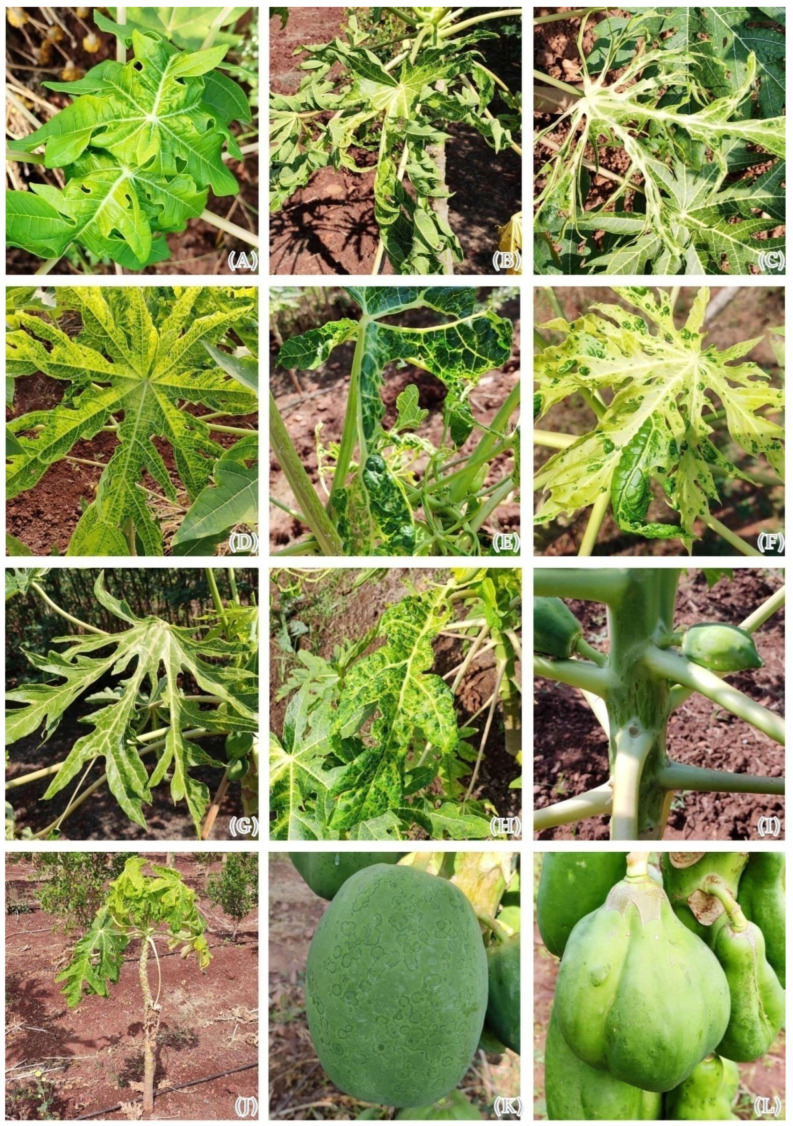
PRSV infected papaya plants showing different symptoms under field conditions. (**A**). Green mosaic, (**B**). Leaf curling, (**C**). Shoestring, (**D**). Yellow mosaic, (**E**). Mottling, (**F**). Blistering on leaves, (**G**). Leaves puckering, (**H**). Leaves distortion, (**I**). Oily streak, (**J**). Stunted growth, (**K**). Ring spot, (**L**). Bumps and distorted fruit.

**Figure 2 pathogens-12-00824-f002:**
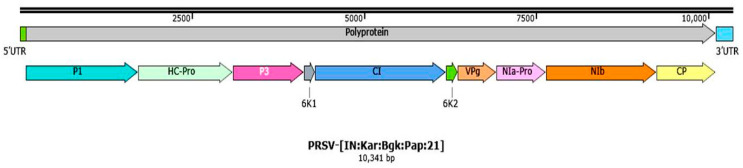
Complete genome organization PRSV-BGK isolate (OL677454). UTR: Untranslated region, P1: Protease, HC-Pro: Helper Component protease, CI: Cylindrical Inclusion, VPg: Viral Protein genome linked, NIa-Pro: Nuclear Inclusion a protease, NIb: Nuclear Inclusion b and CP: Coat Protein.

**Figure 3 pathogens-12-00824-f003:**
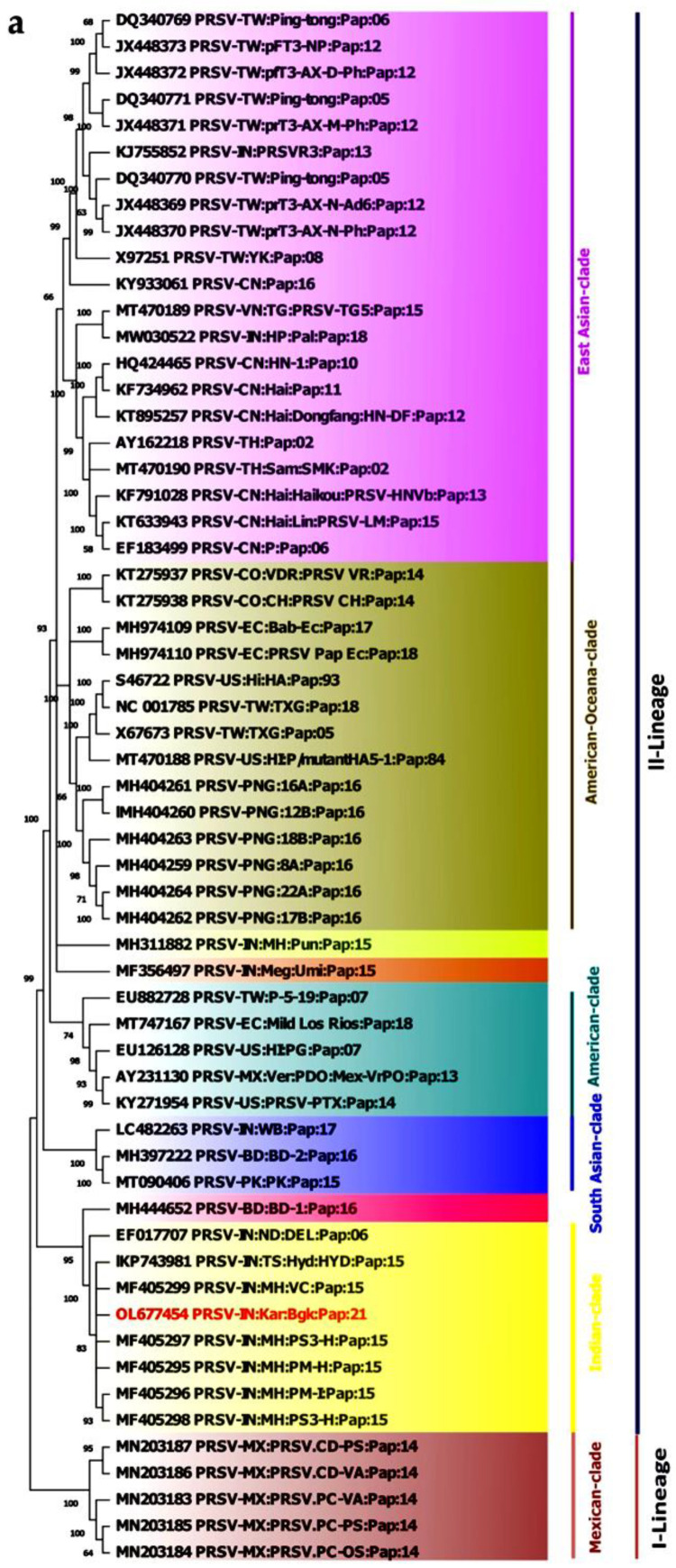
Phylogenic tree showing the relationship of PRSV-BGK isolate (OL677454) with other selected PRSV isolates based on deduced aa sequence of complete genome (number at each node indicates 60 percentage bootstrap value with 1000 replicates) (**a**); and graphical representation of percent pairwise scores for aa identity matrix using SDT (**b**).

**Figure 4 pathogens-12-00824-f004:**
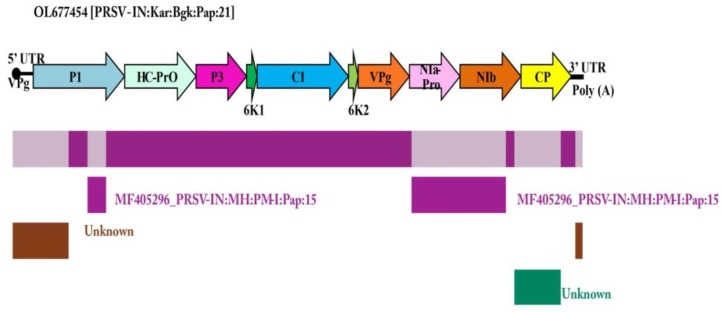
Recombination breakpoint analysis of PRSV-BGK (OL677454) isolate with other PRSV isolates reported across the world. [The recombinant fragments are shown as shaded bars with the origin (parental virus species)].

**Figure 5 pathogens-12-00824-f005:**
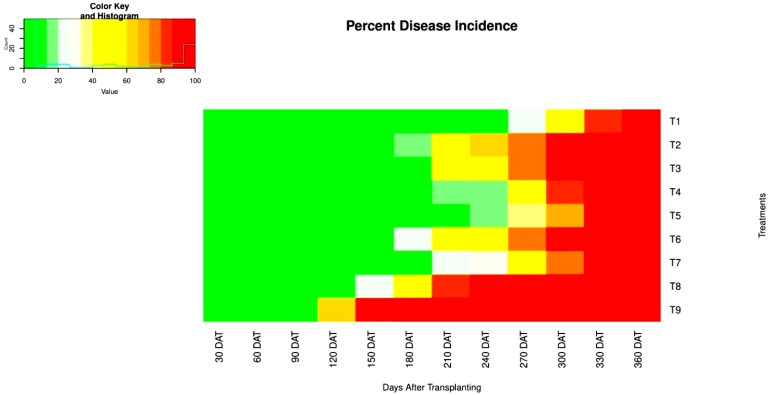
Heatmap generated on percent disease incidence of PRSD at different crop growth stages using Rstudio. (DAT: Days after transplanting; T: Treatments).

**Figure 6 pathogens-12-00824-f006:**
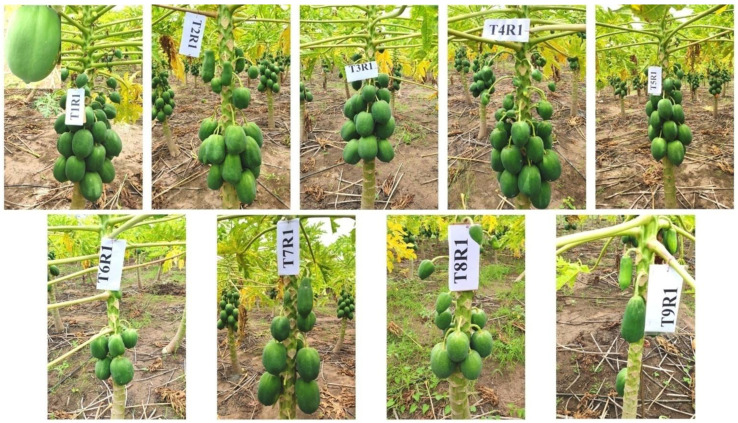
Fruits on papaya plants in different treatments of experiment on integrated management of PRSD.

**Figure 7 pathogens-12-00824-f007:**
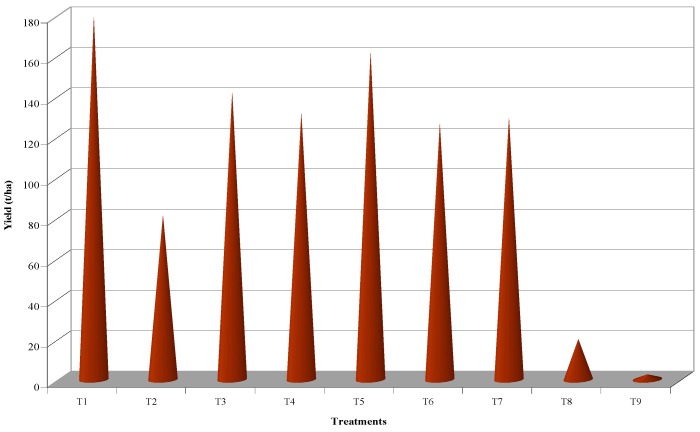
Effect of insecticides and biorationals on yield of papaya.

**Table 1 pathogens-12-00824-t001:** Complete genome organization of PRSV-BGK isolate (OL677454).

Sl. No.	Genomic Regions	Position in Genome (Start-Stop Codon)	Predicted ORF’s Size (nt)	Predicted Size of Protein (No. of Amino Acids)	Predicted Protein Weight (kDa)	Predicted Cleavage Site	Strand Orientation	Direction with Start Codon
1	Genome length (nt)	1–10,341	-	-	-	-	-	-
2	Polyprotein	86–10,114	10,029	3342	380.24	-	+ve	Clockwise
3	5′UTR ^a^	1–85	85	-	-	-	-	-
4	P1 ^a^	86–1723	1638	546	62.3	MEQY/N	+ve	Clockwise
5	HC-Pro ^a^	1724–3094	1371	457	52.12	HYIVG/G	+ve	Clockwise
6	P3	3095–4129	1035	345	39.9	VIHQ/A	+ve	Clockwise
7	6K1	4130–4285	156	52	6.01	VYHQ/S	+ve	Clockwise
8	CI ^a^	4286–6190	1905	635	71.35	VYHQ/G	+ve	Clockwise
9	6K2	6191–6361	171	57	6.36	VFHQ/G	+ve	Clockwise
10	VPg ^a^	6362–6928	567	189	21.33	VHHE/G	+ve	Clockwise
11	NIa-Pro ^a^	6929–7642	714	238	26.51	VFEQ/S	+ve	Clockwise
12	Nib ^a^	7643–9253	1611	537	61.82	VYHQ/S	+ve	Clockwise
13	CP ^a^	9254–10,114	861	286	32.72	-	+ve	Clockwise
14	3′UTR	10,115–10,341	227	-	-	-	-	-
15	Name assigned based on NSI	Papaya ringspot virus-[India:Karnataka:Bagalkote:Papaya:2021]						
16	Virus acronyms	PRSV-[IN:Kar:Bgk:Pap:21]						
17	Accession no.	OL677454						

^a^ UTR: Untranslated region, P1: Protease, HC-Pro: Helper Component protease, CI: Cylindrical Inclusion, VPg: Viral Protein genome linked, NIa-Pro: Nuclear Inclusion a protease, NIb: Nuclear Inclusion b and CP: Coat Protein.

**Table 2 pathogens-12-00824-t002:** Recombination breakpoint analysis of PRSV-BGK isolate and its putative parental sequences of PRSV-BGK isolate.

Virus	Events	Breakpoint (nt)		Region/s	Recombination Parents		*p*-Value						
		Begin	End		Major Parent	Minor Parent	RDP	Geneconv	BootScan	MaxChi	Chimaera	SiScan	3Seq
**PRSV-[IN:KAR:BGK:Pap:21] OL677454**	1	1	1031	5′ UTR to P1	PRSV-[EC:Mild-Los-Rios:Pap:18] MT747167	PRSV-[IN:Meg:Umi:Pap:15] MF356497	3.251 × 10^−6^	- *	-	1.140 × 10^−7^	5.526 × 10^−9^	1.026 × 10^−6^	6.155 × 10^−12^
		10,328	10,341	3′ UTR									
	2	1373	1710	P1	PRSV-[IN:TS:Hyd:HYD:Pap:15] KP743981	PRSV-[IN:MH:PM-I:Pap:15]MF405296	3.407 × 10^−2^	-	-	6.918 × 10^−4^	-	-	7.044 × 10^−3^
	3	7327	9061	NIa-Pro to NIb	PRSV-[IN:TS:Hyd:HYD:Pap:15] KP743981	PRSV-[IN:MH:PM-I:Pap:15] MF405296	2.677 × 10^−47^	2.990 × 10^−41^	-	8.308 × 10^−16^	2.461 × 10^−16^	2.373 × 10^−21^	6.155 × 10^−12^
	4	9205	10,075	NIb to CP	PRSV-[TW:TXG:Pap:18] NC001785	PRSV-[IN:PRSVR3:Pap:13] KJ755852	3.041 × 10^−5^	-	-	5.194 × 10^−5^	1.131 × 10^−5^	5.576 × 10^−3^	4.050 × 10^−8^

* Nonsignificant value.

**Table 3 pathogens-12-00824-t003:** Effect of insecticides and biorationals on PRSD incidence (pooled analysis of two seasons data recorded during 2019–2020 and 2020–2021).

Disease Incidence (%) at												
Treatments	30 DAT *	60 DAT	90 DAT	120 DAT	150 DAT	180 DAT	210 DAT	240 DAT	270 DAT	300 DAT	330 DAT	360 DAT
T_1_	0.00	0.00	0.00	0.00	0.00	0.00	1.49	5.19	24.07	50.90	84.25	100
	(0.00) **	(0.00)	(0.00)	(0.00)	(0.00)	(0.00)	(4.75)	(10.79)	(29.31)	(45.59)	(70.89)	(90.00)
T_2_	0.00	0.00	0.00	0.00	4.00	19.65	54.15	61.56	79.39	92.82	100	100
	(0.00)	(0.00)	(0.00)	(0.00)	(8.21)	(25.99)	(47.44)	(51.93)	(67.49)	(78.99)	(90.00)	(90.00)
T_3_	0.00	0.00	0.00	0.00	0.00	9.58	43.04	47.79	74.07	89.33	100	100
	(0.00)	(0.00)	(0.00)	(0.00)	(0.00)	(16.23)	(40.61)	(43.63)	(64.50)	(74.28)	(90.00)	(90.00)
T_4_	0.00	0.00	0.00	0.00	0.00	2.61	15.38	19.54	46.28	81.57	100	100
	(0.00)	(0.00)	(0.00)	(0.00)	(0.00)	(5.42)	(20.92)	(25.90)	(42.83)	(65.18)	(90.00)	(90.00)
T_5_	0.00	0.00	0.00	0.00	0.00	0.00	10.97	13.75	39.81	71.88	100	100
	(0.00)	(0.00)	(0.00)	(0.00)	(0.00)	(0.00)	(17.56)	(21.55)	(38.37)	(58.34)	(90.00)	(90.00)
T_6_	0.00	0.00	0.00	0.00	3.65	25.10	46.04	48.94	74.29	90.50	100	100
	(0.00)	(0.00)	(0.00)	(0.00)	(7.83)	(29.99)	(42.72)	(44.39)	(59.79)	(77.35)	(90.00)	(90.00)
T_7_	0.00	0.00	0.00	0.00	0.00	2.19	23.60	28.22	52.31	75.56	100	100
	(0.00)	(0.00)	(0.00)	(0.00)	(0.00)	(5.54)	(28.09)	(31.71)	(46.81)	(60.65)	(90.00)	(90.00)
T_8_	0.00	0.00	0.00	10.63	25.56	55.56	81.47	87.03	100	100	100	100
	(0.00)	(0.00)	(0.00)	(17.19)	(30.18)	(48.38)	(68.84)	(74.80)	(90.00)	(90.00)	(90.00)	(90.00)
T_9_	0.00	0.00	0.57	65.61	90.38	100	100	100	100	100	100	100
	(0.00)	(0.00)	(2.49)	(54.19)	(73.76)	(90.00)	(90.00)	(90.00)	(90.00)	(90.00)	(90.00)	(90.00)
S Em ±	NS	NS	0.53	1.43	2.22	2.35	3.85	3.04	5.13	3.48	2.17	NS
C D at 5%	NS	NS	1.51	4.09	6.35	6.72	11.02	8.68	14.66	9.94	6.21	NS

* DAT: Days after transplanting, ** Figures in parentheses are arc sine transformed values.

**Table 4 pathogens-12-00824-t004:** Effect of insecticides and biorationals on growth, yield, and yield parameters of papaya influenced by PRSD (pooled analysis of 2019–2020 and 2020–2021).

Treatments	Plant Height (cm) at 270 DAT *	Internodal Length (cm) at 270 DAT	Plant Girth (cm) at 270 DAT	No. of Leaves per Plants at 270 DAT	No. of Days Taken for First Flowering	No. of Flowers at 270 DAT	No. of Days Taken from Flowering to First Fruit Set	No. of Days Taken to First Fruit Set to Harvest	No. of Fruits per Plant	Fruit Diameter (cm)	Fruit Length (cm)	Fruit Breadth (cm)	Fruit Cavity Diameter (cm)	Fruit Yield per Plant (Kg)	Fruit Yield per Hectare (t)
T_1_	225.93	4.11	40.45	29.50	93.01	73.06	94.01	125.64	26.09	13.13	27.78	14.43	6.81	57.86	178.56
T_2_	204.98	3.60	37.44	23.19	105.33	32.05	103.90	110.44	16.22	9.89	20.25	9.93	6.07	26.18	80.79
T_3_	206.43	3.71	38.26	27.19	100.83	58.26	101.07	118.44	23.03	11.43	24.75	11.36	6.53	45.83	141.44
T_4_	214.18	4.10	39.48	28.29	99.31	56.70	95.93	123.50	22.43	12.05	24.69	12.18	6.62	42.45	131.01
T_5_	216.71	4.10	39.68	28.67	95.94	59.88	96.38	123.88	24.21	12.78	25.42	12.55	6.73	52.16	160.97
T_6_	205.79	3.49	37.75	26.35	101.93	54.40	100.85	114.42	21.69	10.82	22.93	11.04	6.40	40.76	125.80
T_7_	214.49	3.73	39.12	27.31	100.19	56.67	101.43	115.97	21.75	11.21	24.39	11.56	6.45	41.75	128.83
T_8_	160.95	3.39	34.05	20.54	105.47	20.73	111.15	98.33	5.66	8.88	17.78	9.46	5.94	6.43	19.84
T_9_	135.53	3.28	20.95	19.80	106.81	15.15	115.18	98.58	1.19	8.53	15.06	8.91	5.63	0.81	2.51
S Em ±	1.79	0.04	0.18	0.21	0.17	0.70	0.46	0.46	0.54	0.13	0.19	0.10	0.05	0.79	2.45
C D at 5%	5.11	0.11	0.51	0.59	0.49	2.00	1.31	1.32	1.54	0.36	0.54	0.30	0.14	2.27	7.01

* DAT: Days after transplanting.

## Data Availability

Sequence of papaya ringspot virus (PRSV) isolated from papaya in this study was submitted under accession no. OL677454.

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
