# Peer review of "Survey, Detection, Characterization of Papaya Ringspot Virus from Southern India and Management of Papaya Ringspot Disease"

_pathogens, 2023, doi:10.3390/pathogens12060824_

Round 1

Reviewer 1 Report

-In abstract and introduction:

-Line 18: PRSV is not a species, use papaya ringspot virus instead Papaya ring spot virus

-Please use italic format for approved virus species in the abstract and other parts of the paper. A virus name should never be italicized, even when it includes the name of a host species or genus, and should be written in lower case. This ensures that it is distinguishable from a species name, which otherwise might be identical. The first letters of words in a virus name, including the first word, should only begin with a capital when these words are proper nouns.
- write ringspot instead of ring spot

-Line 19: Review the ICTV format for virus classification.

I suggest:

This virus belongs to the species Papaya ringspot virus, genus Potyvirus, Family Potyviridae.

In abstract, lines 39-42:

-A treatment comprising eight sprays of four different insecticides and micronutrients at 30-day intervals was observed to be the best treatment in managing the disease, with no incidence of PRSD up to 180 days after transplanting (DAT).

-This variable (PRSD incidence) was analyzed visually or by PCR. Although PRSV induces clear symptoms of the presence of the disease, should be consider that the disease could be present in the plants and not manifest visible symptoms, this could be associated with the cultural measures applied, and the use of micronutrients.

-Line 67:

Although papaya ringspot is the most prevalent disease worldwide, leaf curl diseases is not prevalent in the America, specify then.

-Line 95: The name ‘Papaya ringspot virus’ was coined….

It is suggested: The name ‘papaya ringspot virus’ was coined…

-Line 115: prevalence of the disease, characterization

Include ´molecular´ characterization because not were carried out studies of transmission, hosts, etc..

-Line 186: The reference of ´the agronomical 186 practices as per the recommended package of practices (POP) of the UHS´ are available? Cite the reference.

Comments:

In the introduction, the chronological and updated history of the disease and its management were discussed. Minor details were suggested.

Materials and Methods

Line 136: To confirm the PRSV in papaya samples / instead To confirm PRSV in ´infected´ papaya samples

-In the epigraph 2.2, validation/verification, test to detect other viruses could be considered?

Results

-In Table S5, what Pathogenicity test did you used?

-In line 224 to 228 was described the PRSD symptoms, I am considered that symptoms of Figure 1 B,F,J could be associated to mixed infections which PRSV and other viruses. Under field conditions would recommend testing for the presence/absence of other viruses implicated.

-In line 234: The authors refer that ´Distortion of leaves was seen only in those locations where PRSD incidence was more 234 than 90.0 percent´. The severity It is mainly associated with several factors, climate, time of infection, crop variety, isolates, but not incidence. Why this observation was referred?

-Line 236: In epigraph 3.2 it is suggested to include ´Mollecular´ characterization because in this epigraph only molecular aspects were included.

Line 400-402: Include some sentences of these results.

-In the discussion Epigraph, data that are not part of the results of this work are discussed ie. Line 589-602

General conclusions

According to results, the combination of four different insecticides and micronutrients was the most effective treatment for reducing the incidence of PRSD under field conditions. In the Integrated disease management modules for the management of PRSD, module M1 was designed by looking into the treatment T1, the most effective treatment in the above study that consists of 12 foliar sprays of four different insecticides followed by micronutrients. Others practices were not mentioned.

In the last decades was postulated by several authors that insecticide sprays against the aphid vectors are not effective in reducing virus disease because aphids transmit virus before the insecticides act to kill them (Webb and Linda, 2007), considering that these insecticides are of fast mechanism of action, other complementary practices could be addressed or discussed (crop barriers, protection of the nursery with polypropylene mesh). The establishment of crop barriers (Zea mays L., sugarcane, etc) has been evaluated in the control of PRSV-P in various countries. The interest in include other practices as an integrated disease management tool is the result of concerns about potential negative effects of pesticides on human health and the environment, resistance, etc. In general, combining insecticidal control, micronutrients, cultural practices, and trap crop can increase the effectiveness of PRSD management in India. If these practices are included in the recommended package of practices (POP) of the UHS, Bagalkot should be referred in this research.

Papaya ringspot virus (PRSV) is a major limiting factor in papaya production throughout the tropics and subtropics. Manuscript pathogens-2377886 (Survey, detection, characterization of Papaya ringspot virus from Southern India and Management of Papaya Ringspot disease) described epidemiology and management studies of PRSD in districts of Karnataka state, India. Moreover, was evaluated the effect of various treatments for the management of PRSD under field conditions. The research is interesting and thoroughly conducted. The manuscript is also well written although a few minor changes should be considered to improve its overall quality. 

Author Response

Please see attachement for Answers to Reviewer-1 Comments

Reviewer 2 Report

Papaya ringspot virus is described from southern India. This reviewer notes a paper published 5 years ago with a similer title: Pushpa, R.N.; Shantamma; Anil, P.; B, M.; Bhose, S.; Sawan, K.; Rangaswamy, K.T.; Girish, T.R.; Nagaraju, N. Molecular Characterization, Epidemiology and Management of the Papaya Ringspot Virus (PRSV) in Papaya under Southern Indian Conditions. Int. J. Agric. Sci. 2018, 0975-3710.

The authors must familiarise themselves with virus nomenclature, as advised by the ICTV here https://ictv.global/faq/names/ The authors have written species names, not virus names, including in the title. Species do not infect plants, viruses do. The species name cannot be amended to PRSV. A ‘PRSV’ isolate is not a member of a genus, it is a member of a species. Please amend throughout the manuscript.

Clarify ‘roving survey’. This is a term I am unfamiliar with.

L94-98. Please clarify if the cause of PRSD is PRSV?

L103 states ‘However, at present, no variety of papaya is showing absolute resistance to PRSV.’ There are a number of reports that state otherwise, notably:

Jia, R., Zhao, H., Huang, J., Kong, H., Zhang, Y., Guo, J., Huang, Q., Guo, Y., Wei, Q., Zuo, J. and Zhu, Y.J., 2017. Use of RNAi technology to develop a PRSV-resistant transgenic papaya. Scientific Reports, 7(1), p.12636.

Mendoza, E.M.T., Laurena, A.C. and Botella, J.R., 2008. Recent advances in the development of transgenic papaya technology. Biotechnology annual review, 14, pp.423-462.

Chen, G., Ye, C.M., Huang, J.C., Yu, M. and Li, B.J., 2001. Cloning of the papaya ringspot virus (PRSV) replicase gene and generation of PRSV-resistant papayas through the introduction of the PRSV replicase gene. Plant Cell Reports, 20, pp.272-277.

Fang, J., Lin, A., Qiu, W., Cai, H., Umar, M., Chen, R. and Ming, R., 2016. Transcriptome profiling revealed stress-induced and disease resistance genes up-regulated in PRSV-resistant transgenic papaya. Frontiers in Plant Science, 7, p.855.

L110 reads ‘Therefore, an alternative to pesticide use of biorationals and biostimulant (seaweed) based management methods have gained importance in recent decades [31-39]. I could see little about virus control in any of these references

L112: ‘few attempts have been made to evaluate biorationals and biostimulants as management practices against insect pests in controlling viral diseases [40]’ This reference is from a very weak journal and appears to be mainly about the chemical composition of bovine urine. I see nothing about controlling plant viruses.

I know of no scientific basis for using neem oil, pongamia oil, ground-nut oil, mineral oil, seaweed extract and micronutrients to control PRSV. If there is a basis for these treatments, you have clearly stated it.

OK

Author Response

Please see the attachment for Answers to Reviewer-2 comments

Reviewer 3 Report

I have read your paper “Survey, detection, characterization of Papaya ringspot virus from Southern India and Management of Papaya Ringspot disease”, which provides detailed information about the PRSV (characterization and comparison with other isolates, etc) and PRSD (incidence, symptoms, etc) on papaya crop in India. Also, the work explores the potential treatment for the control of PRSD considering agronomical interest of this crop (as yield, costs, benefits, etc.).   

Some revisions were made.

General revisions:

-          Please see Zerbini et al., 2022, in order to write propertly the name of a virus and specie (Zerbini, F.M., Siddell, S.G., Mushegian, A.R. et al. Differentiating between viruses and virus species by writing their names correctly. Arch Virol 167, 1231–1234 (2022). https://doi.org/10.1007/s00705-021-05323-4 )

. Virus names should not be italicized, even when they include the name of a host species or genus. Correct the virus name on title, line 18, Keywords (write it as papaya ringspot virus)

-          Why there is not information about prevalence on results? since in methodology it is mentioned the prevalence determination.

-          The presence of other viruses on the samples was analysed?. There is any possibility of coinfection?  Could you add this information to the discussion section?

-          I suggest to discuss why two season were considering in field experiments instead of three, as generally recommended.   

1.       Abstract

-          Line 28 – 30. Please rewrite the sentence to clarify it and add the identity values (%) at  aa and nt level.

-          I suggest improving the abstract in order to highlights the results according to the methodology implemented in this work and to  (there are a lot of works done by the authors and I think they should be reflected in the abstract)

2.       Introduction

-          Line 68: replace “disease” by “virus”

-          Line 107: eliminate “and insecticides” since insecticides are considered pesticides

-          Line 110: improve the writing … for example “Therefore, an alternative to pesticide use are the biorationals and biostimulant (seaweed) based management methods, which have gained importance in recent decades”

3.       Materials and Methods

Line 118 (Survey and collection of virus isolates): Mention how the prevalence was determined.

-          Line 135. Summarize the subtitle (for exemple “Detection and sequencing of PRSV”)

-          Line 138. Eliminate “Quality of”

4.       Results

-          Line 304-308 “Pairwise sequence identity analysis was carried out using the Sequence Demarcation Tool (SDT) version 1.2 with default parameters to obtain the color coded matrix of the identity scores. The nt and aa identity of the complete genome sequence of PRSV-BGK isolate (OL677454) was compared with the complete genome sequences of PRSV isolates reported in the GenBank from different parts of the world.: These sentences should be moved to methodology.

-          Line 309: correct “hade” for “had a”

Author Response

Please see the attachement for Answers to Reviewer 3 comments

Round 2

Reviewer 2 Report

I remain unconvinced that this work represents significant scientific advances despite some improvements to the manuscript after review.

OK
